# Seroprevalence and factors associated with hepatitis B virus exposure in the incarcerated population from southern Brazil

Joelma Goetz de Gois[1]☯, Sthefanny Josephine Klein Ottoni Guedes[2]☯, Ana Paula Vieira[1]☯, Franciele Aní Caovilla Follador[1]☯, Luís Fernando Dip[3]☯, Léia Carolina Lucio[3]☯, Kérley Braga Pereira Bento Casaril[3]☯, Paulo Cezar Nunes Fortes[3]☯, Valdir Spada Júnior[3]☯, Guilherme Welter Wendt[3]☯, Lirane Elize Defante Ferreto[1,3]☯ *

1 Health Sciences Center, Postgraduate Program in Applied Health Sciences, Western Paraná State University, Francisco Beltrão, Brazil, 2 Health Sciences Center, Faculty of Medicine, Public Health Lab, Western Paraná State University, Cascavel, Brazil, 3 Health Sciences Center, Faculty of Medicine, Public Health Lab, Western Paraná State University, Francisco Beltrão, Brazil

☯ These authors contributed equally to this work.
* liraneferreto@uol.com.br

**Data Availability Statement:** Data cannot be shared publicly because of the National Health Council Resolution number 466 (http://conselho.

## Abstract

Hepatitis B viral infection (HBV) in prisons poses serious public health challenges because it significantly contributes to the increase in both morbidity and mortality indicators worldwide. Research has shown high HBV prevalence among inmates when compared to the general population. In this study, we estimated the prevalence of HBV exposure and its risk factors among 1,132 inmates detained in high security institutions. A cross-sectional, epidemiological study was carried out in 11 male-only prisons in the State of Paraná, Brazil, between May 2015 to December 2016. HBV exposure was explored using a variety of methods, including HBsAg, anti-HBs, and total anti-HBc. Data were analyzed using univariate and multivariate techniques. The overall prevalence of HBV exposure was 11.9% (95% CI: 10.9–12.8), totaling 135 individuals. In the multivariate analyses, risk factors that remained statistically significant were related to the penitentiary location (Francisco Beltrão; OR = 5.59; 95% CI: 3.32–9.42), age (over 30 years; OR = 5.78; 95% CI: 3.58–9.34), undergoing tattooing procedures in prison (OR = 1.64; 95% CI: 1.03–2.60), self-reported sexual activities with a known drug user (OR = 1.67; 95% CI: 1.12–2.48) and having a history of previous history of hepatitis B or C infection (OR = 2.62; 95% CI: 1.48–4.64). The findings indicate that public policies–including vaccination, early diagnosis, harm reduction strategies, and adequate treatment–should be designed and delivered in the same way for both the incarcerated and the general population in order to reduce the prevalence of HBV and its associated consequences.

## Introduction

Viral hepatitis is a major cause of morbidity and mortality around the globe [1]. Although vaccination campaigns for hepatitis B virus (HBV) infection reduced the prevalence and

saude.gov.br/resolucoes/2012/Reso466.pdf), - items "II.25", "III 1 q", and "IV.3 e". The resolution states that researchers could solely use and share the material and data obtained in the research exclusively according to the consent of the participants and/or partner institutions. At the time of data collection, participants were not asked to consent that their data could be shared. As such, any further use and/or sharing of data must be approved beforehand by contacting the Western Paraná State University Institutional Ethics Committee via e-mail (cep.prppg@unioeste.br) or telephone +(5545 3220-3056).

**Funding:** albeit we did receive funding for the research (Brazilian Ministry of Health [Ministério da Saúde -BR; grant 797322/2013], the funders had no role in study design, data collection and analysis, decision to publish, or preparation of the manuscript.

**Competing interests:** The authors have declared that no competing interests exist.

socioeconomic impact of the virus in industrialized countries, evidence obtained via systematic review suggest that about 257 million people are currently living with HBV [2].

Among the numerous infectious diseases, hepatitis B is one of the diseases that contribute significantly to the increase in human morbidity and mortality indicators [3] as well as to elevated socioeconomic impact [4]. It is estimated that 5% of the world population carries HBV; furthermore, approximately one million individuals die from chronic liver disease every year [5, 6].

Prevalence of HBV remain high and in vulnerable groups such as incarcerated populations. Indeed, it is estimated that nearly 10 million individuals are incarcerate worldwide and that about half-million of those (or 4.8% of the global prison population) may be infected with HBV [7].

Research in the Brazilian prison population has shown great variability in the prevalence of HBV, ranging from 3.8% to 54.5% [7–11]. Incarceration increases the risk of HBV infection, especially when associated with structural characteristics of prisons and with the marginalized social position occupied by the population deprived of liberty [5, 9, 12].

The prevalence of HBV as measured by HBsAg in inmates is also highly variable elsewhere. For instance, prevalence and West and Central Africa can be as high as 23.5%; in Eastern Europe and Central Asia, data indicates to a 10.4% prevalence [7]. Lastly, a 5.7% prevalence of HBV is found in East and Southern Africa prisons (5.7%) [7]. These findings are worrying, considering that immunization effectively prevents HBV infection, thus protecting against disability, physical and psychological disorders, and death [13, 14].

Unprotected sexual intercourse, syringe sharing, inadequate access to health services, and tattooing in inappropriate places are important predictors of HBV transmission [15]. Many times, individuals are already admitted to the prison unit carrying the virus [16], and the prison system may act as a hotspot for virus dispersion beyond the incarcerated population [17–19].

According to the 2020 National Survey of Penitentiary Information, Brazil has 702,069 people deprived of liberty in all regimes [20]. Although there are health care policies for this population, practices are still quite incipient. It is reasonable to imply that both the lack of financial and human resources and the lack of understanding regarding the importance of health policies targeting this vulnerable group might favor the spread of infectious diseases in prisons [21]. Thus, this investigation sought to explore the seroprevalence and risk factors associated with HBV exposure among male inmates.

## Methods

### Study design and setting

The present study is part of a cross-sectional survey conducted from May 2015 to December 2016 in 11 penitentiaries located in the State of Paraná, Brazil. Based on information from the Paraná Penitentiary Department (DEPEN/PR), at the time of the study, there were about 19,000 inmates distributed in 23 closed male penitentiary institutions. Of the 23 high security closed prisons, 11 were included in the larger study and are located in six cities in the state (Francisco Beltrão, Londrina, Curitiba, São José dos Pinhais, Pinhais, and Piraquara). For the present study three cities from the six listed were selected. The choice was based on the population size of the city and on the fact that they have closed penitentiaries in the main city and in metropolitan cities, where the inmate remains every day in the prison unit under supervision.

### Participants

Proportional stratified sampling was conducted using each prison as a randomization unit. A total of 1,132 inmates were interviewed with 60 losses or refusals (5%). The sample size was calculated based on an expected prevalence of 50% HBV seroprevalence with a 1% variation, 80%

power and 5% alpha error. The total population (8,142 inmates) thus resulted on a minimum sample size of 942 individuals. We added approximately 25% individuals (1,192 inmates) to account possible losses due to refusals to participate in the research.

## Procedures

This study was conducted with the approval of the Ethics Committee on Human Research of the Western Paraná State University, under opinion number 810.574. All eligible participants provided written informed consent prior to participation on a voluntary basis and no compensation was provided. The treatment offered to individuals who did or did not participate in the study was the same.

On the day of data collection, inmates were sorted numerically in ascending order from lists provided by the prison. A list of random numbers was generated using Microsoft Excel® 2007 software. To be eligible to participate, individuals had to be (a) 18 years or older, (b) be formally under state custody, (c) be able to consent for themselves, (d) be suitable to be interviewed by one researcher only, and (e) be able to understand spoken Portuguese.

Participants completed a questionnaire with demographic information, time in prison, illicit drug use, self-reported knowledge regarding previous infection with hepatitis B and C, and sexual behavior. Individuals were also asked about their knowledge of the transmission and evolution of diseases such as HIV, hepatitis B, hepatitis C, and other sexually transmitted infections. Vaccination status was obtained from medical records in the institutional databases.

Blood samples were collected from each participant. After serological separation, the serum was stored at -20°C. Serum samples were used to detect the following serological markers: HBsAg anti-HBs and total anti-HBc.

To investigate the seroprevalence of HBV antibodies, we used the following procedures: HBsAg qualitative analysis (HBsAg Qualitative II kit, Abbott GmbH & Co. KG, Wiesbaden, Germany), anti-HBs (Anti-HBs kit, Abbott GmbH & Co. KG, Wiesbaden, Germany) and anti-HBc (Anti-hepatitis B core antigen antibody II Assay kit, Abbott GmbH & Co. KG, Wiesbaden, Germany) in serum samples on a i4000 instrument (Architect System, United States of America [USA]). Test results were analyzed according to the manufacturer's instructions. The serological results were delivered individually by the researcher to the research participants. Those with results indicating exposure were referred for specialized follow-up with an infectious diseases doctor within the Public Health System.

Data were analyzed using the Statistical Package for the Social Sciences v. 24 (SPSS Inc., Chicago, IL, USA). Chi-square and Fischer's exact tests were conducted to investigate the association between the studied variables. From univariate analyses, results with a p value of .20 or less were selected for inclusion in further multivariate analyses using unconditional logistic regression.

## Results

Participants who presented the following markers were considered to have current or previous HBV infection: HBsAg+ total and Anti-HBc (7 prisoners), Anti-HBc and Anti-HBs (100 prisoners), and Anti-HBs alone (28). Thus, 135 prisoners were positive for one or more markers, representing an overall prevalence of 11.9% (95%CI: 10.0–13.8) (Table 1).

The seroprevalence was approximately three times higher in Francisco Beltrão than in the other places of data collection. In all municipalities, a higher seroprevalence of HBV was observed among people over 30 years old. Only in Francisco Beltrão there was a higher prevalence observed among individuals who reported having no knowledge about HIV.

**Table 1.** Characterization of the sample and seroprevalence of hepatitis B in 11 state penitentiaries in three municipalities of Paraná, Brazil.

| Variable | Francisco Beltrão (n = 119) | Londrina (n = 276) | Curitiba (n = 737) | Total |
|---|---|---|---|---|
| **Total prevalence** | 31.1% | 10.9% | 9.2% | 11.9% |
| | | | | |
| **Age** | p < .001 | p < .001 | p < .001 | p < .001 |
| Between 18 and 30 years old | 15.6% | 2.9% | 2.6% | 4.1% |
| More than 30 years old | 49.1% | 18.7% | 16.3% | 20.2% |
| **Education** | p = .565 | p = .190 | p = .861 | p = .190 |
| Incomplete elementary school | 33.8% | 13.7% | 8.9% | 13.3% |
| Complete elementary school | 27.1% | 8.0% | 9.5% | 10.6% |
| **Was employed when arrested** | p = .346 | p = .604 | p = .555 | p = .468 |
| Yes | 25.5% | 12.0% | 8.4% | 11.1% |
| No | 35.3% | 9.3% | 9.9% | 12.7% |
| **Number of times in the prison system** | p = .074 | p = .584 | p = .639 | p = .042 |
| Once | 36.5% | 12.2% | 8.4% | 14.2% |
| Twice | 17.6% | 9.4% | 9.7% | 10.1% |
| **Conviction time** | p = .378 | p = .673 | p = .903 | p = .392 |
| Up to 10 years | 26.3% | 9.6% | 9.5% | 11.1% |
| More than 10 years | 35.5% | 11.9% | 8.9% | 12.9% |
| **History of sexually transmitted infection** | p = .111 | p = .262 | p = .214 | p = .338 |
| Yes | 19.4% | 14.4% | 11.8% | 13.4% |
| No or does not know | 36.1% | 9.1% | 8.4% | 11.4% |
| **Got a tattoo in prison** | p = .179 | p = .148 | p = .004 | p = .001 |
| Yes | 23.5% | 7.1% | 4.6% | 7.7% |
| No | 36.8% | 13.4% | 11.5% | 14.3% |
| **Got a piercing in prison** | p = .741 | p = 1.000 | p = .863 | p = .563 |
| Yes | 16.7% | 0 [b] | 6.7% | 7.7% |
| No | 31.9% | 11.0 | 9.3% | 12.1% |
| **Shared objects** | p = .601 | p = .885 | p = .345 | p = .340 |
| Yes | 29.3% | 10.5% | 8.5% | 11.3% |
| No | 37.0% | 12.5% | 11.0% | 13.4% |
| **Knowledge about HIV status** | p = .004 | p = .222 | p = .798 | p = .038 |
| Yes | 23.9% | 9.5% | 9.4% | 10.9% |
| No | 55.6% | 16.4% | 8.3% | 16.3% |
| **Knowledge about hepatitis status** | p = .153 | p = .826 | p = 0.174 | p = .771 |
| Yes | 23.6% | 9.6% | 11.1 | 12.4 |
| No | 37.5% | 11.4% | 7.9 | 11.6 |
| **Have participated in a preventive campaign** | p = .245 | p = .829 | p = 0.906 | p = .946 |
| Yes | 50.0% | 7.7% | 8.9% | 12.5% |
| No | 29.0% | 11.2% | 9.3% | 11.9% |
| **Information about hepatitis B transmission and evolution** | p = .600 | p = .555 | p = 0.977 | p = .362 |
| Yes | 32.4% | 11.5% | 9.2% | 12.4% |
| No | 21.4% | 6.3% | 9.2% | 9.6% |
| **Blood transfusion** | p = .599 | p = .863 | p = 0.337 | p = .482 |
| Yes | 44.4% | 11.5% | 12.4 | 13.9 |
| No | 30.0% | 10.7% | 8.8 | 11.6 |
| **Sex with a drug user** | p = .001 | p = .698 | p = .020 | p < .001 |
| Yes | 17.5% | 10.1% | 7.0 | 8.8% |
| No | 46.4% | 12.4% | 12.3 | 16.5% |

(*Continued*)

**Table 1.** (Continued)

| Variable | Francisco Beltrão (n = 119) | Londrina (n = 276) | Curitiba (n = 737) | Total |
|---|---|---|---|---|
| **Has used injection drugs** | p = .932 | p = .973 | p = .838 | p = .982 |
| Yes | 33.3% | 11.1% | 10.5% | 11.3% |
| No | 31.0% | 10.9% | 9.1% | 12.0% |
| **Sexual orientation** | p = .189 | p = .763 | p = .441 | p = .172 |
| Heterosexual | 33.3% | 10.8% | 9.6% | 12.7% |
| Other | 9.1% | 14.3% | 6.3% | 7.1% |
| **Homosexual relationship** | p = .682 | p = .938 | p = .986 | p = .960 |
| Yes | 100%[a] | 13.6% | 9.3% | 12.1% |
| No | 30.5% | 10.6% | 9.2% | 11.9% |
| **Have intimate visitation** | p = 0.134 | p = .215 | p = .345 | p = .342 |
| Yes | 17.9% | 7.9% | 11.0% | 10.5% |
| No | 35.2% | 13.3% | 8.5% | 12.6% |
| **Condom distribution in prison** | p = 0.941 | p = .337 | p = .086 | p = .965 |
| Yes | 29.5% | 9.1% | 11.3% | 12.1% |
| No or does not know | 32.0% | 13.5% | 7.3% | 11.8% |
| **Use of alcoholic beverage** | p = 0.139 | p = .541 | p = .309 | p = .051 |
| Yes | 29.2 | 10.3% | 8.9% | 11.4% |
| No | 66.7 | 16.7% | 15.0% | 20.0% |
| **Ever had hepatitis (B or C)** | p = 0.069 | p = .670 | p = .002 | p < .001 |
| Yes | 58.3 | 16.7% | 21.3% | 25.3% |
| No | 28.0 | 10.5% | 8.1% | 10.8% |
| **Had hepatitis B vaccine** | p = .115 | p = .249 | p = .238 | p = .977 |
| Yes | 40.4% | 6.7% | 6.9% | 11.7% |
| No | 25.0% | 12.4% | 10.1% | 12.0% |

Data are expressed as percentages (%). P values were obtained using the Chi-Squared test.

[a] Only one subject in the cell.

[b] No subjects in the cell.

The main factors associated with hepatitis B were: age, having only one passage through the prison system, getting a tattoo in prison, and having reported any previous history of hepatitis B and C infection. After inserting these variables in the multivariable model, it was observed that inmates from Francisco Beltrão and individuals older than 30 years had an OR higher than five for HBV seroprevalence compared to Curitiba and younger individuals, respectively. The other risk factors that remained statistically significant were having already had hepatitis B or C, not having tattoos, and not using illicit drugs, presenting approximately twice the OR compared to their peers (Table 2).

## Discussion

HBV can cause acute or chronic hepatitis, which could potentially develop into cirrhosis of the liver, liver cancer and, therefore, leading to death. HBV infection can go undetected for many years as the onset of the disease and initial developments of liver damage are often asymptomatic. It is estimated that a high percentage of the population is infected and undiagnosed, which contributes to the spread of the disease and poor prognosis [22]. Moreover, there is a higher risk of HBV transmission in prison settings due to a lack of knowledge about the forms of viral transmission, repeated and unprotected sexual intercourse, high proportion of

**Table 2. Factors associated with hepatitis B seroprevalence in eleven state penitentiaries in three cities of Paraná, Brazil.**

| Independent variable | OR (95% CI) | p | OR$_{AJ}$ (95% CI) | p |
|---|---|---|---|---|
| **Location** | | | | |
| Francisco Beltrão | 4.43 (2.80–7.04) | < .001 | 5.59 (3.32–9.42) | < .001 |
| Londrina | 1.20 (.76–1.89) | .431 | --- | --- |
| Curitiba | 1 | | 1 | |
| **Age** | | | | |
| Between 18 and 30 years old | 1 | | | |
| More than 30 years old | 5.88 (3.72–9.30) | < .001 | 5.78 (3.58–9.34) | < .001 |
| **Education** | | | | |
| Incomplete Elementary School | 1 | | | |
| Complete Elementary School | .77 (.54–1.11) | .161 | --- | --- |
| **Knowledge about HIV status** | | | | |
| Yes | 1 | | | |
| No | 1.59 (1.05–2.41) | .030 | --- | --- |
| **Sexual orientation** | | | | |
| Heterosexual | 1 | .132 | | |
| Other | 1.54 (.25–1.20) | | --- | --- |
| **Use of alcoholic beverage** | | | | |
| Yes | 1 | .034 | | |
| No | 1.94 (1.05–3.60) | | --- | --- |
| **Number of times in the prison system** | | | | |
| Once | 1 | | | |
| Twice | .68 (.47 –.68) | .034 | --- | --- |
| **Got a tattoo in prison** | | | | |
| Yes | 1 | | 1 | |
| No | 2.01 (1.32–3.06) | .001 | 1.64 (1.03–2.60) | .037 |
| **Sex with a drug user** | | | | |
| No | 1 | | 1 | |
| Yes | 2.04 (1.42–2.93) | < .001 | 1.67 (1.12–2.48) | .013 |
| **Ever had hepatitis (B and/or C)** | | | | |
| Yes | 2.81 (1.68–4.68) | < .001 | 2.62 (1.48–4.64) | .001 |
| No | 1 | | 1 | |

OR$_{ADJ}$ (95%CI) values are presented only for the variables that made up the final model.

injecting drug users (IDU), and practices involving tattooing and piercing [23]. Moreover, both social and individual vulnerability suffered by population in prisons can also increase the risk of HBV transmission. After leaving prison, individuals are also at higher risk for HBV infection [7, 14, 24–30]. In this study, the goal was to estimate the prevalence of HBV infection and its risk factors among 1,132 Brazilian inmates, all detained in high security institutions. Consequently, prisons are considered reservoirs of various infectious agents, representing a high-risk environment for the transmission and expansion of emergency/emerged diseases for public health [18, 19].

In the current study, it was observed a prevalence of hepatitis B three times higher in the municipality of Francisco Beltrão when compared to the other municipalities. This finding is consistent with the data from the Health Secretary of Paraná (2018) because the western region of the state ranks higher in the number of hepatitis B cases [29]. In addition, it should be

considered that the coverage and uptake of HBV vaccination varies substantially by geographic region and prison category [30].

Still regarding the municipality of Francisco Beltrão, a higher seroprevalence of HBV was found among individuals who showed no knowledge about HIV. This lack of knowledge is alarming since people living with HIV and people in prison are high risk groups for HBV infections [22]. Thus, it is reasonable to infer that the current educational programs aiming to prevent HBV are insufficient. Comprehensive programs are needed, including particularly components addressing inmates concerns [18]. The Centers for Disease Control and Prevention has highlighted the importance of blood screening for the virus and subsequent anti-HBV immunization in the prison population. representing an opportunity to prevent viral infection in a highly vulnerable, at-risk population [19].

Furthermore, our study found that HBV prevalence was higher among those aged 30 years or more. For the municipality of Francisco Beltrão, this chance increased more than five times for hepatitis B compared to the capital (Curitiba) and in comparison to younger inmates. In the literature, few studies are available that have reported age-specific prevalence for HBV [31]. Nonetheless, a study conducted in Mato Grosso do Sul, Brazil, reported a significant, positive association of anti-HBc prevalence and age [9]. In this respect, age may be understood as a variable of lifetime exposure, thus indicating cumulative risks for HBV infection [9, 32]. Another important point that may be related to age is the absence of vaccines until 1998, when three-dose hepatitis B immunization became available in the public health network.

A significant relationship between previous incarceration history and HBV infection was also found. This association has been mentioned previously, albeit the mechanisms are not yet fully explained [33, 34]. In this sense. further research is needed to clarify the relationship between arrest and anti-HBc positivity [34].

Regarding IDU, the present study found a higher prevalence of HBV exposure in individuals who do not inject drugs. Indeed, the most reported form of HBV transmission is sexual, followed by vertical transmission [35]. Vertical transmission is a frequent cause of HBV spread in regions of high endemicity. In the state of Paraná, higher number of cases are found in the western and southwestern regions, comprising the regions of Pato Branco, Francisco Beltrão, Foz do Iguaçu, Cascavel, and Toledo [29]. Nonetheless, the higher prevalence of HBV infection found in individuals who do not inject drugs might seem contradictory given that inmates are at higher risk of contracting blood-borne viruses (BBV) as a result of high levels of involvement with IDU [32]. However, a cohort investigation conducted in Sri Lanka found no relationship between these infections and intravenous drug use [36]. In addition, a recent Scottish study showed very promising results in which HBV vaccination was associated with reduced chances of infection among IDU [37].

Similarly, there is evidence that tattooing is one the key risk factors for blood-borne diseases. In the prison environment, tattoos are performed using non-sterilized equipment that are also shared between inmates [38]. Although this study found lower prevalence of HBV in individuals who do not have tattoos, it is essential to pursue universal immunization against hepatitis B and support educational activities focusing on inmates' health in conjunction with rigorous, safer tattooing practices in prison to prevent hepatitis B transmission [39].

In summary, data gathered in the current investigation indicates that inmates are at high risk of HBV infection due to social and environmental risk factors, such as intravenous drug use, unprotected sex, tattooing, restricted space, and overcrowding. Thus, preventive vaccination efforts can be both a challenge and an opportunity. Individuals who would not have access to health services in the community can be reached and followed for a period of time during incarceration [2, 19, 40].

The influx of people between prisons and the community—staff and inmates—means that undiagnosed and untreated infections in prisons reflect and amplify the prevalence of infection in surrounding communities [7, 18]. This extent varies across settings depending on incarceration rates, criminal justice responses to drug use and sexual behavior in the community and in prison, the prevalence of infection, and risky behaviors in prison settings [41]. It must be emphasized that treatment of chronic hepatitis B is highly effective and leads to viral suppression in 90% of cases [42]. Thus, accelerated hepatitis B immunization schedules could result in rapid seroconversion for short-term, early protection for susceptible patients. Furthermore, collaboration should be sought among public health officials, physicians and correctional authorities to implement screening and vaccination programs in individuals at high risk for HBV infection to limit the spread of the disease [19].

Finally, it seems important to highlight that this study contains some important limitations, including the nature of its design that impeded the establishment of casual links between the variables analyzed. Additionally, the lack of both HBV DNA and HBeAg testing represents another weakness of the study, which might encourage future researchers to address the issue in order to advance the understanding of the prevalence of HBV in prisons and its associated risk factors.

## Conclusions

The data from this study demonstrate that there is a high seroprevalence of HBV exposure in the prison environment, particularly among inmates presenting with factors associated with hepatitis B virus exposure. Furthermore, there is evidence that current efforts to prevent infectious diseases in prisons are insufficient. Hence, it is essential to expand testing procedures to identify inmates in the need of professional care. Educational programs and widespread immunization campaigns are needed to contain the spread hepatitis B virus within prisons.

## Author Contributions

**Conceptualization:** Sthefanny Josephine Klein Ottoni Guedes, Luís Fernando Dip, Léia Carolina Lucio, Kérley Braga Pereira Bento Casaril, Paulo Cezar Nunes Fortes, Valdir Spada Júnior, Lirane Elize Defante Ferreto.

**Data curation:** Guilherme Welter Wendt, Lirane Elize Defante Ferreto.

**Formal analysis:** Luís Fernando Dip, Léia Carolina Lucio, Kérley Braga Pereira Bento Casaril, Paulo Cezar Nunes Fortes, Valdir Spada Júnior, Guilherme Welter Wendt, Lirane Elize Defante Ferreto.

**Investigation:** Ana Paula Vieira, Franciele Aní Caovilla Follador.

**Methodology:** Joelma Goetz de Gois, Ana Paula Vieira, Franciele Aní Caovilla Follador, Luís Fernando Dip, Léia Carolina Lucio, Kérley Braga Pereira Bento Casaril, Paulo Cezar Nunes Fortes, Valdir Spada Júnior, Guilherme Welter Wendt, Lirane Elize Defante Ferreto.

**Project administration:** Lirane Elize Defante Ferreto.

**Writing – original draft:** Sthefanny Josephine Klein Ottoni Guedes, Luís Fernando Dip, Léia Carolina Lucio, Kérley Braga Pereira Bento Casaril, Paulo Cezar Nunes Fortes, Valdir Spada Júnior.

**Writing – review & editing:** Joelma Goetz de Gois, Sthefanny Josephine Klein Ottoni Guedes, Ana Paula Vieira, Franciele Aní Caovilla Follador, Luís Fernando Dip, Léia Carolina Lucio,

Kérley Braga Pereira Bento Casaril, Paulo Cezar Nunes Fortes, Valdir Spada Júnior, Guilherme Welter Wendt, Lirane Elize Defante Ferreto.

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
