## [Decision Letter · Decision Letter 0]

5 Jun 2022

PONE-D-22-06352Seroprevalence and factors associated with hepatitis B virus infection in incarcerated population from southern BrazilPLOS ONE

Dear Dr. Ferreto,

Thank you for submitting your manuscript to PLOS ONE. After careful consideration, we feel that it has merit but does not fully meet PLOS ONE’s publication criteria as it currently stands. Therefore, we invite you to submit a revised version of the manuscript that addresses the points raised during the review process.

We look forward to receiving your revised manuscript.

Kind regards,

Jason T. Blackard, PhD

Academic Editor

PLOS ONE

Journal Requirements:

Additional Editor Comments (if provided):

This is a cross-sectional study of HBV in incarcerated persons in southern Brazil.

Prison populations are poorly studied but likely contribute a significant burden of HBV; thus, studies such as this are relevant.  The overall prevalence was high at 11.9%, although other findings are expected, and the methods are quite standard for an epidemiologic study.

There are multiple awkward phrases.  The manuscript should be reviewed carefully by a native English speaker and/or a professional editing service.

Line 136:  AIDS should be listed as HIV.

Line 158:  does “having already had hepatitis” mean that they knew they had HBV already, had HBV previously and clearly it and now have HBV again, or had HCV previously/currently?  The meaning must be clarified here and elsewhere in the manuscript.

For Tables 1 and 2:  How was knowledge about HIV or hepatitis assesses?  No information is provided in the methods.

In Tables 1 and 2:  what does “ever had hepatitis” refer to?  HBV or HCV or both?

How was HBV vaccination status determined?

Limitations are not listed but include the lack of HBV DNA testing and lack of HBeAg testing.

Reviewers' comments:

Reviewer's Responses to Questions

**Comments to the Author**

1. Is the manuscript technically sound, and do the data support the conclusions?

Reviewer #1: Yes

Reviewer #2: Yes

2. Has the statistical analysis been performed appropriately and rigorously? 

Reviewer #1: Yes

Reviewer #2: I Don't Know

3. Have the authors made all data underlying the findings in their manuscript fully available?

Reviewer #1: Yes

Reviewer #2: No

4. Is the manuscript presented in an intelligible fashion and written in standard English?

Reviewer #1: Yes

Reviewer #2: Yes

5. Review Comments to the Author

Reviewer #1: - The authors should have a look at the statistical analysis package versions that they have included, they differ in the same paragraph Line 145 and Line 148.

- Furthermore, Table 1: The authors should indicate that they have reported percentages

Reviewer #2: The authors determined HBsAg, anti-HBs and total anti-HBc in incarcerated individuals in three settings with the intention of determining the seroprevalence and risk factors. The data presented however does not seem complete because only HBsAg seroprevalence is reported but there is no comment on anti-HBs and total anti-HBc. Also, numbers enrolled in the three sites needs a bit more detail as to how they were enrolled (What was the total number in each setting). The outcomes of the statistical analysis in the tales 1 is confusing as a p-value has been assigned to each column. Were the three columns being compared? Don't we need just one p-value? Hepatitis B infection seems to be defined as the presence of HBsAg so this should reflect in the entire manuscript and must be distinguished from the disease e.g. lines 199-120. The authors must make reference to the tables as they describe the results.

The authors must also shorten the discussion

Below are a few minor comments:

1. Line 81: sentence ends with "and"

2. Line 98: delete sentence beginning with "it is.........."

3. Line 108 following: just make reference to the study (delete the title quoted)

4. Line 133: what intervention are the authors referring to?

5. Line 138: was it serum or plasma or both?

6. Lines 140-143: the authors need to present the name of the assay, the company, the city and country in which it was manufactured

7. Line 141: which lab are the authors refering to? Either they delete or provide the name of the lab.

8. Line 148: Provide the company, the city and country in which it was manufactured

9. Line 262: Delete, "At this juncture"

6. PLOS authors have the option to publish the peer review history of their article (what does this mean?). If published, this will include your full peer review and any attached files.

Reviewer #1: No

Reviewer #2: No

---

## [Author Response · Author response to Decision Letter 0]

4 Sep 2022

The editorial team required one action from our part, namely: to draft a short summary of the restrictions regarding data sharing (it can be about a paragraph long). We included this information in the last submission. Hence, it remains unclear to us if Plos One wants this information to be included in the article or if this information must appear in the Data Availability Statement. Anyhow, we perfected the previous summary explaining the reasons behind the restrictions for sharing this dataset. We also added this information under the “Data Availability Statement” section in the online submission system. The new summary now reads as:

Data cannot be shared publicly because of the National Health Council Resolution number 466 (http://conselho.saude.gov.br/resolucoes/2012/Reso466.pdf), - items “II.25”, “III 1 q”, and “IV.3 e”. The resolution states that researchers could solely use and share the material and data obtained in the research exclusively according to the consent of the participants and/or partner institutions. At the time of data collection, participants were not asked to consent that their data could be shared. As such, any further use and/or sharing of data must be approved beforehand by contacting the Western Paraná State University Institutional Ethics Committee (e-mail cep.prppg@unioeste.br; Telephone +5545 3220-3056).

---

## [Decision Letter · Decision Letter 1]

19 Oct 2022

PONE-D-22-06352R1Seroprevalence and factors associated with hepatitis B virus infection in the incarcerated population from southern BrazilPLOS ONE

Dear Dr. Defante Ferreto,

Thank you for submitting your manuscript to PLOS ONE. After careful consideration, we feel that it has merit but does not fully meet PLOS ONE’s publication criteria as it currently stands. Therefore, we invite you to submit a revised version of the manuscript that addresses the points raised during the review process.

Please make the minor revisions suggested by both reviewers prior to acceptance of your manuscript.

We look forward to receiving your revised manuscript.

Kind regards,

Jason T. Blackard, PhD

Academic Editor

PLOS ONE

Journal Requirements:

Additional Editor Comments:

Please make the minor revisions suggested by both reviewers prior to acceptance of your manuscript.

Reviewers' comments:

Reviewer's Responses to Questions

**Comments to the Author**

1. If the authors have adequately addressed your comments raised in a previous round of review and you feel that this manuscript is now acceptable for publication, you may indicate that here to bypass the “Comments to the Author” section, enter your conflict of interest statement in the “Confidential to Editor” section, and submit your "Accept" recommendation.

Reviewer #1: All comments have been addressed

Reviewer #2: All comments have been addressed

2. Is the manuscript technically sound, and do the data support the conclusions?

Reviewer #1: Yes

Reviewer #2: Yes

3. Has the statistical analysis been performed appropriately and rigorously? 

Reviewer #1: Yes

Reviewer #2: Yes

4. Have the authors made all data underlying the findings in their manuscript fully available?

Reviewer #1: Yes

Reviewer #2: Yes

5. Is the manuscript presented in an intelligible fashion and written in standard English?

Reviewer #1: Yes

Reviewer #2: Yes

6. Review Comments to the Author

Reviewer #1: The authors have satisfactorily addressed all the reviewer's comments and no further revisions. The editor should correct the typographical error on the abstract -Line 10 because it looks like a word is missing. Otherwise, the manuscript should be accepted for publication in PlosOne.

Reviewer #2: Few concerns:

1. The introduction is too long. Lines 90-115, 116-127 may be shortened and portions used while discussing the results. For examples, lines 260 to 266 seems to be a repetition of part of 104-115.

2. I think a better title will be "exposure" not "infection" with HBV. anyone with anti-HBs has recovered so cannot be infected. (see also line 182; antigens were also tested for)

3. Line 250 HBV includes virus (saying HBV virus is therefore a repetition)

7. PLOS authors have the option to publish the peer review history of their article (what does this mean?). If published, this will include your full peer review and any attached files.

Reviewer #1: No

Reviewer #2: **Yes: **Professor Kwamena William Sagoe

---

## [Author Response · Author response to Decision Letter 1]

4 Nov 2022

We are pleased to send you the requested information regarding the “PONE-D-22-06352” in this new cover letter. In summary, we were asked to perform minor reviews that are detailed below. We would like to anticipate that all the suggestions were accepted.

1st request (Editor). 

Response to the 1st request. 

Dear Dr. Jason, we revised all the references. There were no retracted papers cited in our study, only two references in which there were errors regarding authorship naming (i.e., Refence number 14 and Reference number 32) and one correction (errata) for Reference number 35. 

As references 14 and 32 were already cited correctly, we kept these in the text. However, considering that reference number 35 contained substantial corrections, it is somewhat “old” and does not add significant evidence to our manuscript, we decided to exclude it.

We adjusted the citations in our text, and you can clearly see the changes in the file “Revised Manuscript with Track Changes”. Hyperlinks were all checked and minor corrections were performed in the reference list.

For your information, here are the reasons given by the authors for their corrections (ref. 14 and 32) and for the errata (ref. 35).

Reference 14:

February 27, 2019: Minor Correction: Fu Hsiung Su's affiliations of “5, 7, 8, 9” should have included affiliation 3 (Department of Family Medicine, Taipei Medical University Hospital, Taipei City, Taiwan). In the Funding and Grant Disclosures, “Taipei Medical Hospital” should have been “Taipei Medical University Hospital”.

Link: https://peerj.com/articles/4297/

Reference 32:

Correction added on 2 April 2019, after first online publication: This article has been updated to correct the name of the author from ‘John Kaldo’ to ‘John Kaldor’. The second line of the ‘Results’ section of the Abstract has also been corrected to read ‘Hepatitis B’ instead of ‘Hepatitis C’, as indicated by the symbol ᶺ.

Link: https://onlinelibrary.wiley.com/doi/full/10.1111/1753-6405.12870

Reference 35 (excluded in the revised manuscript):

In the MMWR Recommendations and Reports, "Prevention and Control of Infections with Hepatitis Viruses in Correctional Settings," published on January 24, 2003, an error occurred on page 4 in the second sentence of the paragraph under Occupational Exposures. The sentence should read, "Occupational transmission of HBV infection among hospital-based workers has been linked to percutaneous and mucous membrane exposures, and HCV infection has been primarily associated with percutaneous exposure." On page 12, in Box 6, the fourth item under Type of Exposure should read, "Household (e.g., cell or dormitory) contact --- to person with chronic HBV infection." On page 2, errors occurred in Table 1, and on page 20, errors occurred in Table 5. 

Link: https://www.cdc.gov/mmwr/preview/mmwrhtml/mm5210a9.htm

2nd request (Reviewer #1). 

The authors have satisfactorily addressed all the reviewer's comments and no further revisions. The editor should correct the typographical error on the abstract -Line 10 because it looks like a word is missing. Otherwise, the manuscript should be accepted for publication in PlosOne. 

Response to the 2nd request. We thank the reviewer for this remark. Indeed, “HBV infection” was missing from the text. We corrected this issue.

3rd request (Reviewer #2). 

1. The introduction is too long. Lines 90-115, 116-127 may be shortened and portions used while discussing the results. For examples, lines 260 to 266 seems to be a repetition of part of 104-115.

2. I think a better title will be "exposure" not "infection" with HBV. anyone with anti-HBs has recovered so cannot be infected. (see also line 182; antigens were also tested for)

3. Line 250 HBV includes virus (saying HBV virus is therefore a repetition). 

Response to the 3rd request. All these comments are extremely helpful. In the revised file, the reviewer and the editor will note that we: 1) shortened the introduction while also using some portions in the discussion; 2) revised the title and, when appropriated, substituted the word “infection” for exposure in the main text; and 3) deleted “virus” after HBV, hence avoiding repetition. All these changes are clearly marked using the “track changes” option and correspond to the sections of the text mentioned by the reviewer 2.

Finally, we followed the final instructions and removed our previous manuscript files and uploaded only this new Cover Letter. If you need any further information, please, do let us know. Once again, many thanks for your assistance. 

Yours faithfully,

The authors

---

## [Editor Report · Decision Letter 2]

9 Nov 2022

Seroprevalence and factors associated with hepatitis B virus exposure in the incarcerated population from southern Brazil

PONE-D-22-06352R2

Dear Dr. Ferreto,

We’re pleased to inform you that your manuscript has been judged scientifically suitable for publication and will be formally accepted for publication once it meets all outstanding technical requirements.

Kind regards,

Jason T. Blackard, PhD

Academic Editor

PLOS ONE

Additional Editor Comments (optional):

None

Reviewers' comments:

None

---

## [Editor Report · Acceptance letter]

14 Nov 2022

PONE-D-22-06352R2 

Seroprevalence and factors associated with hepatitis B virus exposure in the incarcerated population from southern Brazil 

Dear Dr. Defante Ferreto:

I'm pleased to inform you that your manuscript has been deemed suitable for publication in PLOS ONE. Congratulations! Your manuscript is now with our production department. 

Kind regards, 

on behalf of

Dr. Jason T. Blackard 

Academic Editor

PLOS ONE